# Examining the Availability and Accessibility of Rehabilitation Services in a Rural District of South Africa: A Mixed-Methods Study

**DOI:** 10.3390/ijerph18094692

**Published:** 2021-04-28

**Authors:** Qhayiya Magaqa, Proochista Ariana, Sarah Polack

**Affiliations:** 1Centre for Tropical Medicine and Global Health, University of Oxford, Oxford OX1 3SY, UK; proochista.ariana@ndm.ox.ac.uk; 2International Centre for Evidence in Disability, London School of Hygiene and Tropical Medicine, London WC1E 7HT, UK; sarah.polack@lshtm.ac.uk

**Keywords:** rehabilitation, accessibility, assistive devices, assistive technology, disability

## Abstract

Introduction: Rehabilitation services aim to optimise individuals’ functioning and reduce disability. However, people with disabilities, who represent a key population of users of rehabilitation services, continue to have unmet needs for rehabilitation services that include the provision of assistive devices. This paper examines the availability and accessibility of rehabilitation services in a rural district of South Africa in order to explore why unmet needs for rehabilitation services persist. Methods: All nine district hospitals in a rural district of South Africa were included in the study. Rehabilitation services capacity was assessed by examining the available assistive devices, consumables and human resources at the level of the health facility. Data collection was conducted using the Global Co-operative Assistive Technology [GATE] Assistive Products List, AT2030’s ATScale priority list and the South African National Catalogue of Commodities for Primary Health Care Facilities. Descriptive statistics were then used for the analysis. For the qualitative component, semi-structured interviews were conducted with adults with physical disabilities at household level to explore barriers to accessing assistive device inclusive rehabilitation services and the consequences thereof in the same rural district. An interview guide based on the WHO health system building blocks was used. Thematic content analysis guided the analysis of the interview transcripts. Findings: The findings of the research demonstrate that rehabilitation service capacity in the district was constrained as a result of low availability of assistive devices [2–22%] and consumables [2–47%], as well as, possibly, a shortage of rehabilitation providers [*n* = 30] with an unequal distribution across health facilities [*n* = 9]. In addition, people with physical disabilities reported poor referral pathways, financial constraints, transport and road consideration and equipment unavailability as barriers to accessing rehabilitation services. Moreover, these barriers to access predisposed individuals to finance-, health- and person-related harm. Conclusion: Rehabilitation service availability is constrained by a lack of service capacity in rural South Africa. In addition, the rehabilitation services in district hospitals are not adequately accessible because of existing barriers to enable key populations to achieve optimised functioning.

## 1. Introduction

Rehabilitation services are health services whose chief aim is to optimise individuals’ functioning and reduce disability [1]. These services include, amongst others, physiotherapy, occupational therapy, speech and language therapy, orthotics and prosthetics and audiology services. In addition to having individual benefits for users, rehabilitation services have also been shown to result in cost savings greater than initial investments [2]. Moreover, the growing need for rehabilitation services across countries means that providing rehabilitation services will be a key strategy if the health-related Sustainable Development Goals are to be achieved [1,3,4,5]. However, recent estimates suggest that as many as 2.4 billion individuals globally have unmet needs for rehabilitation services, representing one in three people globally [6]. While rehabilitation services are intended to be used by any individual who has limitations in their functioning, a key population are those with disabilities. People with disabilities continue to have high unmet needs for rehabilitation services, including for assistive devices, in South Africa and globally [7,8,9,10,11,12]. The collective evidence necessitates an inquiry into why individuals, in particular people with disabilities continue to have unmet needs for rehabilitation services despite the benefit of these services for individuals and countries.

Therefore, this paper aims to examine the availability of rehabilitation services in district hospitals of a rural district of South Africa and assess what factors may be influencing access to rehabilitation services amongst people with physical disabilities in the same district. The first objective is to determine the rehabilitation capacity in health facilities according to the elements of assistive devices, consumables and rehabilitation workforce. The second objective is to describe the barriers to access and their consequences for people with physical disabilities.

## 2. Materials and Methods

### 2.1. Study Context

OR Tambo District is a largely rural district in the province of the Eastern Cape in South Africa. The district has a population 1.4 million, making it the district with the highest population in the Eastern Cape [13]. Most of the population lives in widely dispersed homesteads and villages and participates in a subsistence economy [14]. Three-quarters of these households receive social welfare grants intended to alleviate poverty [13].

According to recent health estimates, the largest contribution to mortality in the district is attributable to non-communicable diseases (46%), followed by HIV and TB (32%) [15]. In comparison, South Africa’s leading causes of death in 2019 were HIV/AIDS followed by ischemic heart disease and stroke [16]. Additionally, the Eastern Cape’s prevalence of disability in 2011 and 2016 was 8.6% and 9.6%, respectively, for individuals five years old and older [17]. Regarding access to healthcare, only 4.2% of the population of OR Tambo district has medical scheme coverage, which falls below the coverage of the Eastern Cape (9.8%) and that of the rest of South Africa (15.4%) [15]. Since South Africa has a dual system of private and public health sectors, this means that 96% of OR Tambo district’s population relies on the public health sector for health services including rehabilitation services. The district’s health service delivery platform is formed by a network of 146 primary healthcare facilities (clinics and community health centres), nine district hospitals, two regional hospitals, one provincial central hospital and two private hospitals [15,18].

Regarding rehabilitation services in the district at the primary health care level, services are offered in district hospitals, with outreach services intermittently provided at clinics and community health centres in each district hospital’s catchment area. Recent estimates report that the availability of rehabilitation providers may be low relative to national and global estimates. For instance, per 100,000 population, in 2019, there were 1.5 occupational therapists and 2.1 physiotherapists, which falls below the provincial (2.3 and 2.6, respectively, per 100,000) and national (2.6 and 3, respectively, per 100,000) estimates [15]. Although outdated now, estimates for rehabilitation providers in low-income countries (LICs) and high-income countries (HICs) were 0.5 and 13–16, respectively, per 10,000 population [19].

### 2.2. Capability Approach Framework

The Capability Approach framework by Amartya Sen (Figure 1) is a framework which focuses on the practical opportunities that individuals in a society have available to them in order to live the kind of lives that they consider of value [20]. Therefore, it is concerned with the extent of freedom individuals are afforded in order to pursue and shape the kinds of lives that they desire [21]. The development of this theory was to challenge the convention that more resources equated to better wellbeing. Sen argued that resources undergo conversions before providing a set of practical opportunities that individuals can choose from. These *conversion factors* include age, gender, co-morbidities, impairments, social factors, economic factors or political factors. *Conversion factors* are important because they affect how resources may be translated into a number of capabilities from which individuals may then choose [22]. For example, a bicycle (*resource*) will be of little use for travelling in an individual who has no functioning of their legs (impairments as the *conversion factor*). Similarly, a bicycle will be of no use for travel if the road conditions are bad (environment as the *conversion factor)*. This framework therefore provides an opportunity to examine rehabilitation service availability and its accessibility to a specific population. In this paper, rehabilitation services are the *resource*, while the *conversion factors* are the barriers and facilitators to accessing rehabilitation services. The *choice* element is reflected in the decisions that individuals ultimately make when considering accessing rehabilitation services, resulting in achieved *functionings*. The population of interest is people with physical disabilities in a rural district of South Africa.

While the Capability Approach has largely been applied in the assessment of the unintended outcomes of development interventions, it has since been expanded to other fields including rehabilitation and disability [20,23,24,25]. For instance, Borg et al. applied the framework in examining how assistive devices increase the capabilities of individuals in Bangladesh [26]. In rural South Africa, Sherry [27] applied the framework to demonstrate how disability was a constraining conversion factor when seeking healthcare services. However, these studies tend to focus on discrete elements of the Capability Approach, thus forgoing the analysis of the interactions between components and their effects. This paper addresses the resource (rehabilitation services provided in the public sector), conversion factors (barriers and facilitators to accessing rehabilitation services) and choices of people with physical disabilities as users of rehabilitation services.

### 2.3. Rehabilitation Service Capacity

To examine rehabilitation service capacity at the level of the health facility (district hospital), a selection of elements of rehabilitation services was included as a means to quantify readiness. These elements included assistive devices and consumables used in the provision of rehabilitation services as well as human resources for rehabilitation. Existing health facility assessments such as the service availability and readiness assessment (SARA) and service provision assessment (SPA) do not adequately capture data relevant to assistive devices and rehabilitation consumables [28,29]. In South Africa’s 2013 national baseline health facility audit, the report only indicated whether rehabilitation equipment was available in health facilities but did not specify what types of equipment were assessed [30], therefore making it difficult to compare availability between health facilities and, most importantly, which specific types of equipment were lacking.

As such, the Global Co-operative Assistive Technology (GATE) Assistive Products List (APL) [31] provides the opportunity to assess which assistive devices are available in health facilities and allows comparisons regarding the level of availability across health facilities. The GATE APL contains a list of 50 assistive devices which are relevant for use by individuals with reduced function due to a variety of impairments or due to unaddressed barriers in the environment. The assistive devices relate to mobility, hearing, communication, self-care and vision. The comprehensive nature of the GATE APL, while useful, may make the tool unrealistic because assistive device procurement in countries continues to be constrained by a lack of skilled human resources, funding and prioritisation. Because of this, the ATScale list of five items was also employed to focus the assessment of five priority assistive devices [32]. The ATScale list’s five items, namely wheelchairs, hearing aids, prosthetics, spectacles and smart products, also appear on the GATE APL.

To examine the availability of consumables used in the provision of rehabilitation services, the South African National Catalogue of Commodities for Primary Health Care Facilities was incorporated [33]. The full catalogue specifies all equipment, instruments, consumables and furniture and appliances which should be available for all health services provided in primary health care facilities (Appendix A). Thus, the rehabilitation section for consumables was used in this research. The data for rehabilitation equipment and consumables for each health facility were then entered into a Microsoft Excel spreadsheet for analysis using descriptive statistics for measures of frequency.

### 2.4. Accessibility of Rehabilitation Services

A qualitative inquiry was also pursued to explore what barriers and facilitators exist for individuals residing in the catchment areas of the health facilities (district hospital). Recognising that individuals would have different experiences of seeking rehabilitation services, a constructivist approach [34,35] enabled us to use the experiences of individuals as well as findings from the rehabilitation service capacity assessment to explore how barriers function as conversion factors in accessing rehabilitation services. The population of interest included adults aged 18 years and above residing in OR Tambo district and with a physical disability. Physical disability was chosen to contain the scope of the research in line with resource constraints as well as for pragmatic reasons since physical impairment is relatively easily identifiable by members of the community. Convenience and snowball sampling within the catchment areas of the health facilities was employed [36,37]. First, a list of individuals with physical disability was obtained from a conveniently selected cohort of community leaders. The community leaders were asked to identify individuals who had any deformity, injury or condition (recent or long standing) which affected how an individual moved or walked. Next, additional names were generated through snowballing by the individuals on the list. Individuals were then called and, after a concise description of the research, were asked if they would consent to take part and, if so, provide directions to their homes. At the household, participants were screened for the presence of physical disability using the Washington Group Extended Set questions (mobility module) [38]. If an individual answered *a lot of difficulty* or *cannot do at all* to at least one of the questions they were eligible for inclusion in the study [39]. This threshold is likely to identify moderate to severe forms of disability and may miss milder forms of disability [40]. Capturing milder forms of disability would have substantially increased those who would have qualified as having a disability. The chosen threshold allowed the research question to be answered whilst also ensuring that the study was feasible given the time and resource constraints.

A semi-structured interview was conducted with each eligible participant in their homes after written informed consent was obtained. We clarified that this was part of a research study and that participants could decline inclusion into the study with no penalty to them or their household. The semi-structured interview guide was developed using the WHO health system building blocks as a framework upon which to develop the questions [41] (Appendix A). Interviews varied from 30 min to 1 h. The interviews were audio-recorded and transcribed from isiXhosa to English by a research assistant. The interviews in each included catchment area reached saturation, where no new information was arising. The English transcripts were then organised in NVivo 11 for analysis [42]. We employed thematic content analysis to guide the analysis since the approach allows the exploration of various perspectives whilst allowing comparisons between participants [43,44]. This approach also focused the analysis whilst still providing ample opportunity for unexpected insights to be identified [44]. First, the author familiarised herself with all the transcripts by reading them and returning to the Xhosa audio material if clarification was required. Next, first-order codes for each transcript followed by second-order themes to connect the codes were created. Ethical approval to conduct the study was obtained from Oxford University (OxTREC 513-19), Walter Sisulu University Human Research Committee (042/2019) and the Eastern Cape provincial Department of Health (EC_201907_001).

## 3. Results

### 3.1. Availability of Rehabilitation Services

All nine health facilities (all of which are district hospitals) in OR Tambo district offered rehabilitation services. The types of rehabilitation services offered in each health facility differed (Table 1). All health facilities offered physiotherapy services. Occupational therapy was available in six of the nine health facilities. Speech and language therapy and audiology services were available in only two of the nine health facilities. No orthotics and prosthetics services were available in the health facilities thus requiring referral to a higher level of care for such services. In only one health facility were all rehabilitation services, except orthotics and prosthetics, offered.

### 3.2. Assistive Devices

Overall, the availability of assistive devices at the level of the health facility was low. The proportions of available assistive devices ranged from 2% to 22% (Figure 2).

Mobility-related assistive devices (Figure 3) were the most frequently available compared to those related to self-care, hearing and communication. Eight health facilities had at least one mobility-related assistive device while five had at least one assistive device or product related to self-care. Only two health facilities had at least one hearing-related assistive device. The most commonly available mobility-related assistive devices were devices walking frames, manual wheelchairs, standing frames and crutches.

Assistive devices availability was low even when examined according to the list of five priority items (Table 2). Wheelchairs were available in five of the nine health facilities and hearing aids were available in the two health facilities which employed audiologists. For spectacles, prosthetics and smart products, patients would need to be referred to higher levels of care or outside the health system.

### 3.3. Consumables

Similar to assistive devices, the availability of consumables across health facilities was low and ranged from 2% to 47% (Figure 4). Consumables used in the provision of physiotherapy and occupational therapy services were the most common, compared to those used in audiology and speech and language therapy services. Ferrules were the most frequently available type of consumable, followed by towelling, exercise bands and wheelchair cushion covers.

### 3.4. Availability of Rehabilitation Providers

In OR Tambo district, there were 30 rehabilitation providers employed in the nine district hospitals (health facilities), all of whom worked in a full-time capacity. By profession, 46% (*n* = 14) were physiotherapists, 33% (*n* = 10) occupational therapists, 7% (*n* = 2) speech and language therapists, 7% (*n* = 2) audiologists and 7% (*n* = 2) physiotherapy assistants (Table 3). The distribution of rehabilitation providers across the health facilities was unequal. The highest number of rehabilitation providers were in HF 3 (*n* = 9), followed by HF 1 (*n* = 5) and HF 2 (*n* = 4). The lowest number of rehabilitation providers were found in HFs 4 (*n* = 1) and 6 (*n* = 1).

The majority (*n* = 17) of the 30 rehabilitation providers employed in the district were community service-level professionals, and they were distributed relatively evenly across all health facilities. They were in their first year of work after university and had not yet obtained independent practitioner status from the Health Professionals Council of South Africa (HPCSA).

### 3.5. Accessibility of Rehabilitation Services

In total, 53 people were screened to determine their inclusion in the study. Five individuals did not meet the screening criteria because they did not report having a *lot of difficulty* or *cannot do at all* for any of the questions. Therefore, 48 adults with physical disabilities from OR Tambo district were included in the study. The characteristics of the included participants are presented in Table 4.

### 3.6. Barriers to Access

#### 3.6.1. Referral Pathways

Some participants reported not being referred for rehabilitation services despite having made contact with a healthcare provider such as a pharmacist, nurse or doctor.


*I only go to the clinic to check up on blood pressure, diabetes and arthritis. I have never been informed about such doctors (physiotherapists). Not even the doctor who referred me to the clinic that I am using.*
*(Female, 75y, HF7)*

Additionally, several participants lamented the inefficient referral pathway to secondary levels of care. One participant observed that the secondary referral hospital which offered orthotic and prosthetics services was always very busy.


*I wish it was not only in (name) hospital where they focus on our type of sickness (impairment) because it gets really full there. All these surrounding hospitals send their patients to that one hospital.*
*(Female, 35y, HF1)*

#### 3.6.2. Financial Considerations

A major theme amongst participants related to financial considerations. Several participants noted limited finances for transport fare as a challenge for going to the health facility. One participant noted having missed his recent physiotherapy appointment at his local health facility as a result and stated:


*I didn’t go because of financial constraints… I take two taxis and same applies too when I am coming back. And the taxi drops me off at (the taxi stop) and I would walk from there to here.*
*(Male, 40y, HF5)*

Financial costs were especially inhibiting for those who needed to hire private transportation because of how remote their households were. It was not uncommon for private car hire costs to the health facility to constitute about a third to half of the month’s household costs.

#### 3.6.3. Transport and Road Considerations

When participants were able to pay for transportation, they experienced challenges with the transportation system itself thus creating additional challenges for accessing rehabilitation services. One participant who mobilises using a wheelchair described their experience of discrimination saying:


*Some of them (taxi drivers) leave you at the side of the road because you are not worth the trouble and some of them tell me that I have to pay for the wheelchair.*
*(Female, 45y, HF5)*

Another participant reported that public transport was scarce where they lived, and, when it was available, it was inaccessible. Therefore, they had to hire a car to go to the health facility.


*They (public transport) are scarce. And taxis are not conducive to my condition anyway*
*(Female, 66y, Hf8)*

#### 3.6.4. Equipment Availability

The availability of equipment at health facilities was another key theme. Several participants described the lack of assistive devices at their relevant health facility as a problem. One participant plainly stated:


*There are no assistive devices for disabled people.*
*(Female, 60y, HF2)*

Another participant noted that they were treated with respect by rehabilitation providers but observed the same challenges with obtaining wheelchairs particularly as it related to the long waiting times.


*I was well-treated. She even told me that it was hard to get wheelchairs. It takes three years for one to get it.*
*(Male, 39y, HF8)*

One participant relayed the stark reality of a health facility that did not have a wheelchair to issue him on the day that he was discharged from hospital saying:


*I was taken out of the hospital on a wheelchair and then put in the car and the wheelchair was taken back to the hospital…There was no explanation given. We were told that the doctor who was supposed to give the wheelchair was on leave at that time.*
*(Female, 70y, HF7)*

Despite resounding agreement amongst most participants about the lack of assistive devices and related consumables in health facilities, there were a few participants who described the availability of equipment as a facilitator to benefitting from rehabilitation services. One participant, who has engaged with rehabilitation services repeatedly noted that the presence of equipment had improved over time, and thus she was able to continue to benefit from rehabilitation services.


*As compared to previous times, now there are machines that can be used to exercise and they go the extra mile to make sure I get what I need.*
*(Female, 45y, HF5)*

Another participant reported that the rehabilitation providers were not only able to provide him with a wheelchair, but that the rehabilitation providers provided one which was suitable for outdoor use on uneven terrain, thus enabling the participant to be able to go outdoors and to push his own wheelchair on even more terrain. He reflected on this improvement saying:


*It was not suitable for outdoor purposes (so) I was given a different wheelchair. There is a difference because the one I used before was not able to go outdoors, it was meant for indoor use only.*
*(Male, 40y, HF3)*

### 3.7. Consequences

The existence of barriers to rehabilitation services for people with physical disabilities were not simply inconveniences, but they also predisposed individuals to harm. For two participants, barriers to accessing rehabilitation services, including assistive devices, at their allocated health facility predisposed them to significant financial losses. For one participant, this was because hiring a private car to attend appointments cost them R450, which constitutes almost a third of the month’s social grant (Female, 75y, HF7). For perspective, one third of the of the income was utilised before groceries and other immediate costs had even been paid for. Another participant reported that she had to purchase a used wheelchair for R500 because she did not have one and did not have other alternatives (Female, 35y, HF1).

In addition to finance-related harm, barriers to rehabilitation services predisposed individuals to health-related harm. While waiting for three months after his accident to receive a wheelchair, another participant reported that she was immobile during that time and “just sat all the time” (Female, 72y, HF7). Prolonged immobility predisposed this participant to an increased risk of obtaining pressure sores. Thus, this waiting period for an assistive device may have compromised her health through developing secondary conditions.

Besides finance-related and health-related harm, participants also encountered personal costs because of barriers to rehabilitation services. One participant reported that, after his first wheelchair that he had obtained from his health facility got old, he tried to obtain a new one by contacting the municipal councillor, who noted his details and requests. However, the promise of a new wheelchair did not materialise, and, as a result, the participant was without a wheelchair for three months. During this period, he had to crawl to the toilet, which is situated outside the house, including when it was raining. Such a situation compromises the dignity of a person.

Another personal cost was expressed by one participant and their primary caregiver. During the interview, both the participant and the participant’s caregiver reflected on their wishes for the participant to recover full or partial mobility functioning because the responsibility of caring for the participant was becoming a burden. Regarding the functioning of the participant, while she could bathe and could change her own incontinence products independently, she was unable to walk and needed to be lifted up from bed to chair. As a result, this participant lived between two homes, that of her mother and that of her caregiver, so as to distribute the care burden.


*When I wake up, Sisi (respectful term for older sister) lifts me up… I wake up and bathe. When I’m done, Sisi puts me on this chair to watch TV and then eat till evening. When it’s time to sleep, Sisi lifts me up again and puts me in bed. She does everything for me.*
*(Female, 27y, HF2)*


*I just wish she could recover and get back to how she was because this is a burden.*
*(Caregiver of Female, 27y, HF2)*

## 4. Discussion

The current study found low availability of AD at the health facility level, with fewer than one quarter of the devices from the GATE APL available across health facilities. This means that the available assistive devices in health facilities were unlikely to meet the full scope of rehabilitation needs since each assistive device needs to cater to the needs of the individual and their environmental circumstances. Previous studies in health facilities in rural South Africa found that only one type of wheelchair, the standard folding frame, was available to be issued to those who needed wheelchairs [45]. Similar to the findings of the current study, this finding suggests that, while wheelchairs may be available in health facilities, there may not be of the variety required to address a diversity of impairments, abilities and environments of each individual patient.

The findings from the current study were also similar to that of a systematic review which found that, overall, access to assistive devices for people with disabilities was low [0–66%] across low- and middle-income countries [46]. However, the studies included in the review utilised a variety of assessment methods including clinical assessment, functional domain self-report and other self-reporting tools. Therefore, comparability of the proportions found in the systematic review and the current study is difficult. Even so, both studies point to a gap in the availability of assistive devices and likely unmet need for assistive devices in South Africa and other LMICs. The GATE APL, which guided the assessment in the current study, is comprehensive and intended to guide countries in formulating their own essential lists of assistive devices [31]. South Africa has not yet formulated its own list of priority assistive devices, but there are guidelines which address various assistive devices, each to varying extents [47,48,49].

A study in an urban and low-income area of South Africa found that unmet need for rehabilitation services, including assistive devices to be about one third of people with disabilities [50]. Another study in South Africa found that only 15.2% of patients who, upon clinical assessment, required hearing aids received them [51]. Both wheelchairs and hearing aids are intended to be available through the public health sector via tender in South Africa. In fact, mobility assistive devices are well represented in South Africa’s National Catalogue for items available on tender, along with hearing and communication assistive devices [33]. These findings of potential unmet need due to limited availability of assistive devices are supported by the findings of the current study. This is despite district hospitals having the largest health expenditure in the provincial health budgets, more so than clinics and community health centres and thus are better resourced to provide rehabilitation services [15]. For example, in OR Tambo district in the 2019 financial year, 26.8% of the provincial budget was allocated to district hospitals, 18.2% was allocated to clinics and 12% was allocated to community health centres. This suggests possible bottlenecks in how items reflected on government tender documents become available in health facilities, thus pointing to challenges in the procurement systems, prioritisation and resource allocation practices at the level of OR Tambo district. For example, limited availability of assistive devices may be related to production factors, as suggested by findings from a study from Tanzania, Malawi and Sierra Leone, which reported the absence of materials and functioning machines required in the production of orthotics and prosthetics [52]. It may also be related to constrained funding procurement systems at the health facility and provincial levels and the lack of population level data on needs in the district [53]. These factors may also explain why the availability of consumables in OR Tambo district was low despite consumables generally being cheaper to procure per item compared to assistive devices. This necessitates an examination of what gaps might exist between South Africa’s rehabilitation policies and the availability of services in health facilities.

Regarding human resources for rehabilitation, in the current study, there were 30 rehabilitation providers employed at district hospitals in OR Tambo district. However, they were not equally distributed across health facilities. Additionally, the distribution of rehabilitation providers according to professional experience was sub-optimal with most rehabilitation departments in OR Tambo district being led by professionals in their first year of work. While it is not known whether the quantity of the 30 providers is adequate to meet the needs, the findings suggest that their distribution in the district’s health facilities is sub-optimal.

Gupta and colleagues’ global assessment of human resources for rehabilitation found that lower income countries had ratios of 0.5 rehabilitation providers per 10,000 population [19]. In contrast, high income countries such as the United Kingdom and Canada had ratios of 13–16 per 10,000 population. In South Africa, the ratios reported in the literature correspond to those found in other LICs, despite South Africa being an upper-middle income country according to World Bank classifications [54]. There are differences in the distribution of rehabilitation providers between higher income and lower income countries. However, this distribution is inequitable because those countries with the highest burden of health conditions and could benefit from rehabilitation services are also the countries which tend to have lower rehabilitation provider to population ratios [19]. Another review found that across country income levels there remained unmet needs for rehabilitation services [55].

Thus, local and global evidence points to a shortage of rehabilitation providers in South Africa and other countries. Based on this literature, the number of available rehabilitation providers in the current study likely means that the number of rehabilitation providers in OR Tambo district is low relative to population-level rehabilitation needs, although this is not certain. This is because the full core rehabilitation team was available in only one of the nine health facilities, two health facilities had only one rehabilitation provider each and most of the rehabilitation workforce consisted of recently qualified providers, meaning that the availability of services was directly dependent on graduate output on a yearly basis. While comparisons may be made within and across countries about the numbers of rehabilitation providers, there is a paucity of literature that establishes thresholds for what an adequate number of rehabilitation providers per population should be [56]. This is also likely related to the lack of population level data on rehabilitation needs. Further research is therefore required to determine acceptable standards for the size of the rehabilitation workforce.

With regards to access, the findings from this study demonstrate that adults with physical disabilities in OR Tambo district experience notable demand and supply-side barriers when accessing public health sector rehabilitation services. This aligns with other research in both HIC and LMIC which shows that, despite being more likely to have poorer health outcomes than those without disabilities, access to rehabilitation for people with disabilities is low [46,57,58]. Similar to the finding of the current study, barriers in the form of poor referral pathways, inaccessible transportation and limited rehabilitation resources are reported in the literature [59,60,61,62].

However, barriers were not simply inconveniences; they exposed individuals and their households to harm related to their finances, health and personal dignity. Since transport was a barrier to accessing rehabilitation services, some participants reported hiring a car from a member of the community, but this was very costly relative to their income. This has been reported in another South African study in which private car hire was reported to cost approximately R500 or more depending on the distance from the health facility [63]. This finding suggests that transportation and financial constraints are both barriers, which also reinforce each other to make accessing rehabilitation services even more complicated. This dynamic and reinforcing interaction between barriers has also been reported in other studies [64]. The implication is that multiple barriers need to be addressed simultaneously in order for access to practically improve.

Moreover, people with disabilities are more likely to be poorer than non-disabled individuals [65]. Financial poverty is not limited to people with disabilities themselves but also extends to the household. For instance, findings from the South African Department of Social Development reported that households in which a person with disability was part of had lower overall household income compared to households which did not have a person with disability [66]. This link between disability and poverty was confirmed by Banks and Polack [67] in their study of LMICs, which showed that the presence of disability was associated with the presence of poverty, and that this relationship was statistically significant across ages and disability types. This means that, when people with disabilities seek rehabilitation and other healthcare, they begin in a position of economic vulnerability. It is therefore not surprising that Hanass-Hancock and colleagues [63] found that people with disabilities encounter economic hardship in the form of both direct (out-of-pocket) and indirect (opportunity) costs when seeking healthcare. A study in India and Cameroon found that individuals with disabilities had to make trade-offs between their own health needs or the needs of the broader family within the household [68].

Drawing from the findings of both the supply-side and demand-side, the Capability Framework presents an opportunity to understand why people with disabilities in OR Tambo district still report unmet needs for rehabilitation services despite these services seemingly being offered in health facilities. The Capability Framework also elucidates the mechanisms which may result in optimal functioning through rehabilitation not being achieved amongst people with physical disabilities in OR Tambo district. First, rehabilitation services are comprised of many components including at the least assistive devices, consumables and rehabilitation providers. What the current study’s findings demonstrate is that a lack of availability of assistive devices and consumables and a possible shortage in rehabilitation providers results in a resource (rehabilitation services) which is deficient and unable to result in desired health outcomes amongst individuals. We were unable to calculate staff to population ratios in OR Tambo district since the study examined only district hospitals and not regional and tertiary hospitals. Second, the presence of barriers [conversion factors] such as inaccessible transportation, lack of finances and absence of assistive devices prevents the ability for accessing rehabilitation services in a way that adequately results in its chief aims of optimising functioning and reducing disability. Lastly, the combination of deficient resources and barriers to access not only reduces opportunities for people with disabilities to achieve what is optimal functioning to them, but this combination also results in finance-, health- and personal-related harm to people with physical disabilities in OR Tambo district. In other words, the reduced access to rehabilitation services as a result of inadequate rehabilitation services accompanied by barriers to access is not only inconvenient but it actively reduces the potential for achieving the kinds of lives that individuals value. Although individuals exercised their agency by hiring private transportation to reach rehabilitation services, the alternative would result in forgoing rehabilitation services. Therefore, while the choices individuals could make were constrained by the barriers (conversion factors), the barriers did not altogether remove the agency of individuals.

A major strength of this study is that all district hospitals in one district were included thus findings may be generalised to the district. A second strength is that both supply-side and demand-side aspects relevant to rehabilitation services were examined, thus enabling the interactions between these aspects to be demonstrated. However, this study also has limitations. First, the assessment of rehabilitation capacity was limited to a selection of readiness indicators to the exclusion of other relevant indicators such as equipment and process related factors such as observed provision of care and guidelines availability. Even so, this study represents an important first step in addressing the gap in rehabilitation capacity assessment in South Africa. Further work is required to develop a health-facility level assessment tool which examines rehabilitation service readiness and processes of care. Second, only people with physical disabilities were included as users of rehabilitation services to the exclusion of other types of disabilities. However, barriers such as finances and equipment are likely relevant for all types of disabilities in the district. Further research should include a diversity of disability types when examining barriers and their consequences. Third, only one investigator conducted the data analysis which may have reduced the rigor somewhat. To mitigate this, the codes and themes were discussed with the advisor as they developed, and the interpretation was discussed with the same advisor. Finally, further work should consider rehabilitation policies and their implementation in order to identify what factors contribute to the available rehabilitation capacity at the health facility and their accessibility to key populations.

## 5. Conclusions

This study showed that the rehabilitation service capacity in a rural district of South Africa is constrained by a low availability of assistive devices, consumables and rehabilitation human resources in health facilities. In addition, people with physical disabilities are hindered by barriers when accessing rehabilitation services. These barriers result from the deficiencies in rehabilitation capacity but also from factors on the demand-side. Implications of this work point to a need for South Africa to develop its own national list of essential assistive products which spans all domains of functioning. Further, additional financing and improved procurement processes for assistive devices and consumables at the health facility level in OR Tambo district are required in order to increase the availability of these items. Finally, there is a need for greater collaboration between the stakeholders involved in social development, transport and roads and health in South Africa in order to address the existing demand-side barriers to rehabilitation services.

## Figures and Tables

**Figure 1 ijerph-18-04692-f001:**
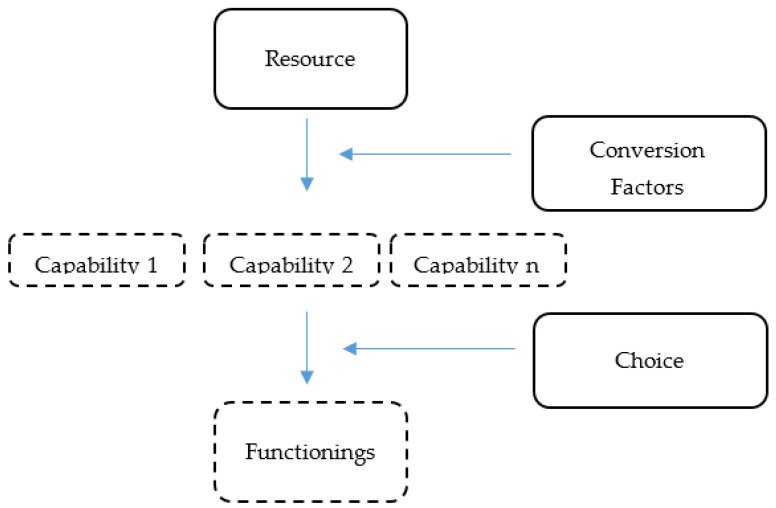
Amartya Sen’s Capability Approach Framework.

**Figure 2 ijerph-18-04692-f002:**
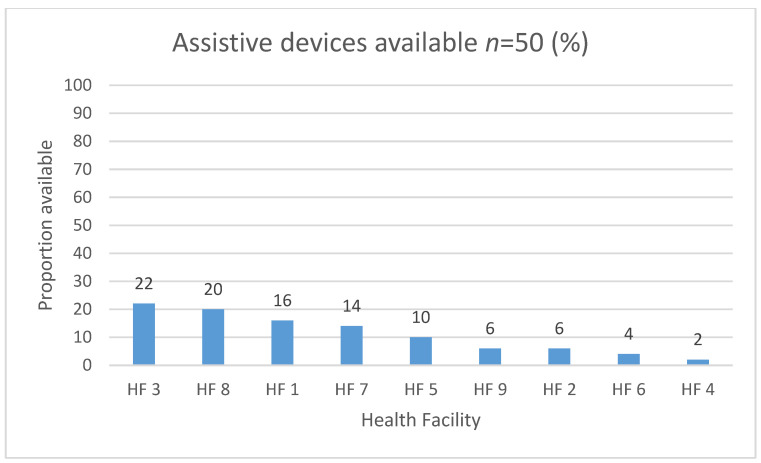
Availability of assistive devices by proportion according to the Global Co-operative Assistive Technology (GATE) Assistive Products List (APL).

**Figure 3 ijerph-18-04692-f003:**
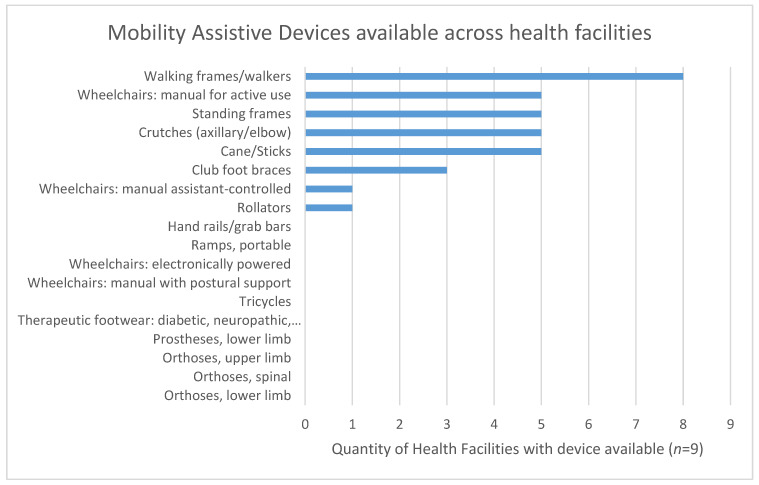
Number of health facilities with availability of mobility-related assistive devices.

**Figure 4 ijerph-18-04692-f004:**
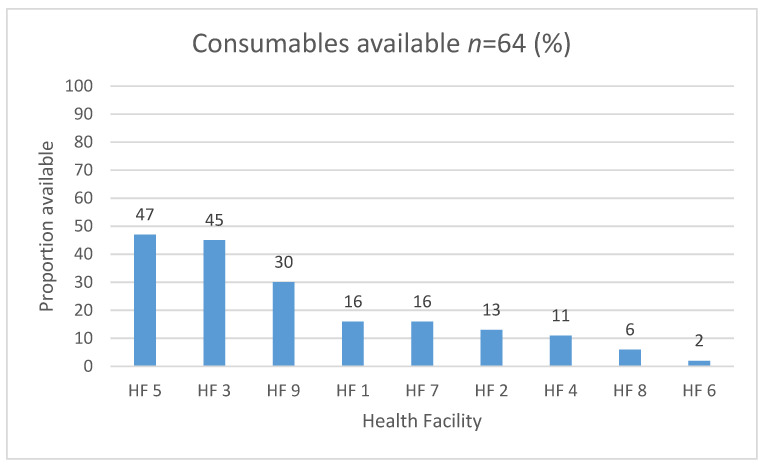
Availability of consumables by proportion in health facilities.

**Table 1 ijerph-18-04692-t001:** Rehabilitation services in OR Tambo district hospitals.

Type of Rehabilitation Service	Number of Health Facilities (*n* = 9)
Physiotherapy	9
Occupational therapy	6
Speech and Language therapy	2
Audiology	2
Orthotics and Prosthetics	0

**Table 2 ijerph-18-04692-t002:** Availability of assistive devices according to ATScale.

	Wheelchairs	Hearing Aids	Spectacles	Prosthetics	Smart Products
Health Facilities (*n* = 9)	5	2	0	0	0

**Table 3 ijerph-18-04692-t003:** Number of rehabilitation providers by profession (full-time posts) in OR Tambo district hospitals.

Profession	HF 1	HF 2	HF 3	HF 4	HF 5	HF 6	HF 7	HF 8	HF 9	TOTAL
Physiotherapists	2	2	4	1	1	1	1	1	1	14
Occupational Therapists	2	2	3	0	1	0	1	0	1	10
Speech & Language Therapists	0	0	1	0	0	0	1	0	0	2
Audiologists	0	0	1	0	0	0	0	0	1	2
Orthotist and Prosthetists	0	0	0	0	0	0	0	0	0	0
Rehabilitation Physicians	0	0	0	0	0	0	0	0	0	0
Rehabilitation Nurses	0	0	0	0	0	0	0	0	0	0
Physiotherapy Assistant	1	0	0	0	0	0	0	1	0	2
TOTAL	5	4	9	1	2	1	3	2	3	30

**Table 4 ijerph-18-04692-t004:** Participant characteristics.

Characteristics	Number (Total = 48)
Gender	Females [21] males [27]
Age	49 years (average) (range 24–83 years)
Education	Incomplete primary [29] Complete primary [4] Incomplete secondary [11] Complete secondary [2] Complete post-secondary [2]
Disability Grant	46 recipients 1 does not qualify 1 in process of application
Assistive device use (combinations possible)	wheelchairs [24] crutches [20] walking sticks [6] orthotics [4] walking frames [3] prosthetic devices [2] no assistive device [1]

## Data Availability

The data presented in this study are available on request from the corresponding author.

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
