# Peer review of "Examining the Availability and Accessibility of Rehabilitation Services in a Rural District of South Africa: A Mixed-Methods Study"

_ijerph, 2021, doi:10.3390/ijerph18094692_

Round 1

Reviewer 1 Report

This manuscript describes a multi-method study of the availability of rehabilitation services and assistive devices and the barriers to accessing these among people with physical disabilities in a rural region of South Africa. The methods and theoretical framework are well-described, the interpretations fit the data, and the paper is very well-written. I have a few minor suggestions that may improve the manuscript:

  1. I believe this would more appropriately be called a multi-method study rather than a mixed-methods study since the quantitative data did not inform the development of the qualitative component (or vice versa). I suggest changing the title to reflect this.
  2. Section 2.4: It would be helpful to add a citation for “a constructivist approach.”
  3. Section 2.4: Please describe how physical disability was defined for the purposes of identifying the list of individual community leaders. In other words, please describe what you asked them for or how you presented the project and potential participants to them.
  4. Section 2.4: Please describe any inclusion/exclusion criteria other than age 18 or older and having a lot of difficulty/can not do at all on one of the questions from the Washington Group. Later, in Section 3.5, you not excluding people because they did not meet criteria but other than these 2 criteria I am not sure why they may have been excluded.
  5. Tables 1 & 2: I recommend adding a % in parentheses next to the numbers.
  6. Figures 2 & 4: I recommend adding the numeric label for the proportion to (above) each of the bars.
  7. Table 3: I recommend changing the title to “Number of rehabilitation providers by profession in OR Tambo district hospitals.”
  8. Section 3.4: Please clarify whether the 17 community-service level professionals are part of the 30 providers referenced in the preceding paragraph or if these are in addition to those 30 independent providers.
  9. Section 4 (Discussion): When you talk about the Capability Framework, it would be helpful to relate back to specific components of the framework. For example, talk about the barriers as conversion factors. Also, please explicitly discuss the role of choice. Was it assessed directly in the interviews (I don’t see it in in the interview guide)? Or is choice irrelevant when the availability of resources is so limited? It would be helpful to address all aspects of the framework here.
  10. Section 5 (Conclusions): An additional limitation I think should be addressed is that it was a single investigator working on the project. While that is understandable, it prevents discussion of codes and themes with others, and so might reduce the rigor somewhat. Please identify this limitation and discuss any additional steps that were taken to address potential bias (e.g., did the research assistants and/or advisors work with you on this or consult in the process?).

Author Response

Dear Reviewer 1

Thank you for your comments. Please see the attachment.

Kind regards

Qhayiya

Reviewer 2 Report

Thank you for inviting me to review the highly relevant and interesting article: Examining the Availability and Accessibility of Rehabilitation Services in a rural District of South Africa: A Mixed-Methods study. The study is well written, the methodology is sound and the topic highly relevant for contextual understanding.

Minor improvements could support the reader to contextualize the findings.

Page 5 paragraph: According to recent health estimates...It would be beneficial for the reader to have a comparison with National data and/or international data regarding prevalence of the reported percentages.

Review figure 2; The graph should be presented on a scale from 0 to 100%It would also facilitate the reader to understand how the percentage regarding availability of services has been calculated; 

Figure 3 the data presented in percentage regarding availability of supports in comparison with the overall sample would understand to estimate the effective proportion and to compare data.

Same for table 2 and Figure 4.

Author Response

Dear Reviewer 2

Thank you for your comments. Please see the attachment.

Kind regards

Qhayiya

Reviewer 3 Report

Thank you for the opportunity to review this paper. This exploration of the use of and access to rehabilitation services in rural South Africa takes a sound mixed-methods approach not only to quantify access issues, but explore in a nuanced way what the outcomes of such issues are, bringing a welcome depth to the paper. The theoretical frameworks used are well chosen, well explained and well employed. The discussion is thoughtful and has clear potential both in practice and policy. With some minor clarifications this paper would make an excellent contribution to the literature:

Abstract: The Methods section of the abstract did not clearly and succinctly describe the quantitative/statistical element of the mixed methods.

  1. Introduction: It might be pertinent here to briefly describe the broader scope of who rehabilitation services serve – in particular, the distinction might need to be made between those who have temporary disabilities as a result of illness or injury (e.g. bad fractures, post-hospital care) and those with total and permanent disability, both from trauma and from birth.

There is also a need to restructure this section (the first paragraph) for flow, as it currently flows from all rehabilitation users to those with disabilities, and then back out again to all those seeking rehabilitation – perhaps move the mention of the narrower focus (those with disabilities) to the end of the paragraph

  1. Materials and Methods: in 2.1, ensure consistency of OR Tambo or O.R. Tambo throughout (just one variation, both are used as is)

2.3 “The data for rehabilitation equipment and consumables for each health facility was then entered into a Microsoft Excel spreadsheet for analysis using descriptive statistics” – what sort of analysis? Need more description of how this analysis was approached and undertaken.

2.4 The threshold for categorization of form of disability between “moderate to severe” and “mild” is clear, but it would help to establish whether this follows a model set by other existing works making a similar distinction, or if this is a novel model of inclusion/exclusion specific to this paper.

It is good that the author specifies a formal ethical approval process was followed and governed the project. It would be additionally fruitful to gain some brief insight into how the researchers also ensured ethical conduct of the research beyond just the signed-form consent process, particularly as the researcher was in the participants’ own home spaces, and their interaction with the participants may have been understood (by participants) as either an additional gatekeeping to the access of rehabilitation services, or as some validation of their rehabilitation access claims that would lead to specific outcomes for them.

3.2 Figure 2 – the differences between HFs may be more exaggerated by the short scale of the Y axis of this graph, as may the true nature of the availability. A better way of displaying these might be vertical percentage bar charts, where each HF has a bar of equal height divided into % availability and % unavailability.

Figure 3 – the figure title does not fit legibly into the figure frame. It might make the figure clearer to remove all the items where quantity = 0, and perhaps list in the text that those items were not available at all.

Figure 4 – as per Figure 2 comment

3.4 Table 3 – clarify if these numbers of providers are counts of individual staff, or full-time equivalent/whole-time equivalent posts. For example, the 4 physios at HF 3 might only work 3 days a week, whereas the 2 at HF1 might both work a full 5-6 day capacity, so a measure of simple staff numbers might not tell the whole story. Total staff counts would be acceptable if FTE/WTE numbers are unavailable; in either case, need to explain/clarify numbers.

Overall: The author should be commended on their good use of inclusive and respectful language throughout (i.e people with disabilities rather than ‘disabled people’).

Written expression is very good, though could benefit from a careful read-through, with a fresh set of eyes (third party), for punctuation use, spelling and grammar.

Author Response

Dear Reviewer 3

Thank you for your comments. Please see the attachment. 

Kind regards

Qhayiya
